# KNOCKOFF-INSPIRED FEATURE SELECTION VIA GENERATIVE MODELS

## ABSTRACT

We propose a feature selection algorithm for supervised learning inspired by the recently introduced knockoff framework for variable selection in statistical regression. While variable selection in statistics aims to distinguish between true and false predictors, feature selection in machine learning aims to reduce the dimensionality of the data while preserving the performance of the learning method. The knockoff framework has attracted significant interest due to its strong control of false discoveries while preserving predictive power. In contrast to the original approach and later variants that assume a given probabilistic model for the variables, our proposed approach relies on data-driven generative models that learn mappings from data space to a parametric space that characterizes the probability distribution of the data. Our approach requires only the availability of mappings from data space to a distribution in parametric space and from parametric space to a distribution in data space; thus, it can be integrated with multiple popular generative models from machine learning. We provide example knockoff designs using a variational autoencoder and a Gaussian process latent variable model. We also propose a knockoff score metric for a softmax classifier that accounts for the contribution of each feature and its knockoff during supervised learning. Experimental results with multiple benchmark datasets for feature selection showcase the advantages of our knockoff designs and the knockoff framework with respect to existing approaches.

## 1 INTRODUCTION

The availability of digital sensors and low-priced storage, computation, and communication channels is becoming increasingly pervasive. As a result, emerging applications involving extremely large datasets are now feasible. Thus, we now often rely on data-centric inference, in which large amounts of data are concentrated and processed via machine learning (ML) algorithms, based on suitable data models, to extract relevant information from the observations. However, such approaches often suffer from high computational complexity as well as limits on available storage, communication, and processing resources. To alleviate such issues, *dimensionality reduction* (Samet, 2006) is commonly applied at the beginning of the data processing pipeline. Dimensionality reduction also addresses multiple issues with the performance of learning and inference from high-dimensional data that are commonly referred to as the *curse of dimensionality* or *Hughes Phenomenon* (Jain & Zongker, 1997; Pavlenko, 2003; Guyon & Elisseeff, 2003).

In the most demanding applications involving real-time data processing or complex sensing architectures (e.g., low-power sensing, streaming data analytics, particle physics, and genetics), it may not be feasible to perform the sensing and/or computation required by most existing approaches to dimensionality reduction during data acquisition. Thus, the practitioner is limited in these cases to *feature selection* (FS) (Guyon & Elisseeff, 2003; Liu & Motoda, 2008), i.e., to selecting the subset of the data dimensions that are most relevant in information extraction. These dimensions are the only ones to be sensed to provide a simple form of dimensionality reduction.

FS has a strong connection with the *variable selection* (VS) problem in regression analysis for statistics (Hastie et al., 2009), where the goal is to find a subset of the predictive variables that capture statistical dependence to the response variable of interest. Informally, the goal of a VS algorithm is to identify as many of the predictive variables on which the response variable is truly dependent (measured by its predictive power) while minimizing the number of selected predictive

variables that are actually unrelated (measured by the false discovery rate); such a strong distinction between features is seldom observed in ML, where the focus of FS is to preserve performance while maximizing dimensionality reduction.

The recent development of *VS via knockoffs* (Barber & Candès, 2015; 2019; Weinstein et al., 2017; Barber et al., 2019) has attracted significant attention in the statistics community due to its guaranteed control of the false discovery rate, as well as its applicability to arbitrary statistical models for the variables. This new framework relies on the creation of samples of *knockoff variables* that provide counterparts to the original variables on existing training samples. The knockoff variable design must meet certain requirements: first, knockoff variables must follow the same probability distribution as the original variables, but they must be independent from the labels to be predicted, conditioned on the original variables; second, they should be pairwise uncorrelated with the original variables in order to provide sufficient predictive power; finally, swapping any two corresponding entries of the original and knockoff sample vectors must not change the likelihoods of these vectors. The framework then relies on numerical scores to compare the statistical dependence of original and knockoff variables with the response variable, discarding those variables for which the scores of these two variable versions (original and knockoff) are indistinguishable. While the original framework focused on jointly Gaussian variable models and Lasso-based linear regression, it has since been extended by new knockoff variable constructions for more sophisticated data models, such as Markov models and Bayesian networks (Sesia et al., 2019; Gimenez et al., 2018; Romano et al., 2019; Fan et al., 2019; Liu & Zheng, 2018; Jordon et al., 2019), as well as proposing additional variable statistics for other regression schemes (Lu et al., 2018); furthermore, technical conditions on the data models and variable statistics have been simplified while preserving the original performance guarantees (Candès et al., 2018).

In this paper, we propose a novel framework for FS in ML inspired by the knockoff framework for VS that is geared towards applications featuring large-scale, high-dimensional datasets. We leverage the flexibility of the framework by designing knockoff feature constructions and scores based on data models that are popular in ML and driven by large-scale datasets. More specifically, we leverage generative models that provide a data distribution characterization that is compatible with the knockoff VS framework. Examples of knockoffs generated according to methods from the literature and from our proposed method are illustrated in Figure 1 for multiple samples from several benchmark image datasets, where it is clearly shown that the properties required for knockoff variables are better met by our constructions than by those previously proposed. It is worth remarking that since the analytical framework for VS performance evaluation does not translate to FS, the proposed knockoff generation methods and scores are best described as heuristics.

This paper's contributions can be summarized as follows. First, we propose a generic algorithm for knockoff variable design that assumes the existence of a generative parameterized data model composed of two mapping functions: one from the data feature space to the space of probability distributions for a low-dimensional parameter, and another from the parameter space to the space of probability distributions for the data. Second, we present example implementations of this algorithm using two standard data-driven models in ML: variational autoencoders and Gaussian process latent variable models. Third, we propose a knockoff-based scoring scheme that is designed for FS for softmax classification. Fourth, we provide numerical examples that compare the performance of the proposed FS schemes against multiple baselines, including approaches based on previously proposed knockoff-based VS designs. Finally, we discuss open questions and directions for future work.

## 2 BACKGROUND

**Feature Selection** (FS) (Guyon & Elisseeff, 2003; Liu & Motoda, 2008) aims to identify a subset $\Omega$ of the indices $\{1, \ldots, N\}$ for the entries of the original feature vector $\mathbf{x} = [x_1, \ldots, x_N]^T \in \mathbb{R}^N$ so that the resulting vector $\mathbf{x}_\Omega \in \mathbb{R}^{|\Omega|}$ best preserves the dataset structure relevant for the ML problem at hand. Three classes of FS approaches have been described in the literature (Guyon & Elisseeff, 2003; Li et al., 2017): filter methods, wrapper methods, and embedded methods. Of particular interest to this paper are *filter methods*, which consider a set of $J$ labeled data examples $\{\mathbf{x}_j, \mathbf{y}_j\}_{j=1}^J$, and compute a score for each dimension $S(n)$ based on the values $x_{j,n}$ and $\mathbf{y}_j$ for each data point. The features are then ranked by this score and those with highest scores compose the set $\Omega$. Common scores include Fisher's prediction score, the data-to-label correlation, the mutual information $I(x_{j,n}; \mathbf{y}_j)$, etc.

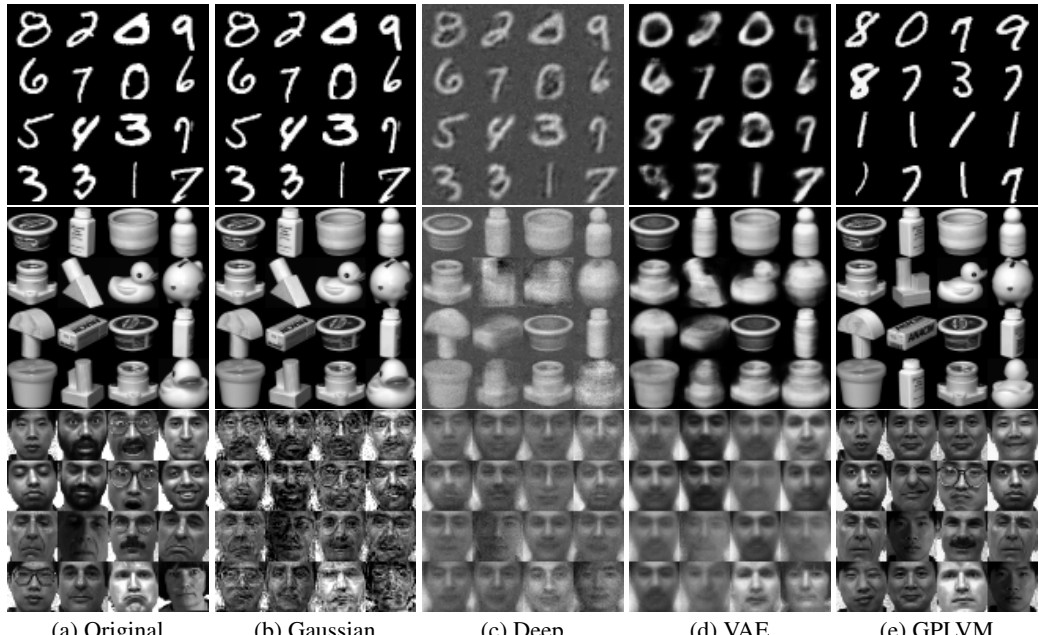

|  (a) Original | (b) Gaussian | (c) Deep | (d) VAE | (e) GPLVM |

Figure 1: *Examples of knockoffs generated from (top to bottom) MNIST, COIL20, and Yale datasets with several data models: (a) Original data, 16 sample images. (b) Multivariate Gaussian model knockoffs (Barber & Candès, 2019), cf. Section 2. (c) Deep knockoffs (Romano et al., 2019). (d) Variational autoencoder (VAE) model knockoffs, Section 3. (e) Gaussian process latent variable model (GPLVM) knockoffs, Section 3.*

**Variable Selection** (VS) in statistics is similar to FS (Hastie et al., 2009). Consider a set of predictive variables $\{x_1, \ldots, x_N\}$ (e.g., the entries of the vector $\mathbf{x}$), and a response variable $y$ (e.g,, the label or parameter to be estimated), and assume that the data points and matching labels correspond to independent and identically distributed (i.i.d.) samples of an underlying joint probability distribution $p(\mathbf{x}, y)$. The goal of VS is to determine the set of *true predictors* $\{x_j\}_{j \in \Omega}$ so that the response is statistically independent of the remaining predictors conditioned on the chosen predictors, i.e., $\mathbf{y} \perp \mathbf{x}_{\Omega^C} | \mathbf{x}_\Omega$; the variables $\mathbf{x}_{\Omega^C}$ are often referred to as *null variables*. The selection mechanism should have high *power* (e.g., high probability of the true predictors $\Omega$ being selected in a set estimate $\hat{\Omega}$) while minimizing the *false discovery rate* (FDR) (e.g., the expected fraction $\mathbb{E}[|\hat{\Omega} \setminus \Omega|/|\Omega|]$ of null variables included within the selected variable set). Standard methods, e.g., (Benjamini & Hochberg, 1995), require knowledge of the conditional distribution $p(\mathbf{x}|y)$.

A recent framework for VS based on the construction of **knockoff variables** (Barber & Candès, 2015; 2019; Weinstein et al., 2017; Barber et al., 2019; Candès et al., 2018) has been shown to control the FDR while being applicable to a wide variety of cases without requiring any knowledge of the conditional response distribution $p(y|\mathbf{x})$. This generality of the approach is given by its single prerequisite knowledge of the distribution of the data prior $p(\mathbf{x})$. The knockoff variables $\tilde{\mathbf{x}} = \{\tilde{x}_1, \ldots, \tilde{x}_N\}$ are constructed for each data sample using this prior so that any swap of a variable with its knockoff does not affect the data distribution. More formally, denote by $(\mathbf{x}, \tilde{\mathbf{x}})_{\text{swap}(\Gamma)}$ the pair of vectors $(\mathbf{x}, \tilde{\mathbf{x}})$ where the entries of each of the vectors corresponding to the indices $\Gamma$ has been swapped with the other. The knockoff variables $\tilde{\mathbf{x}}$ are designed so that $(i)$ $(\mathbf{x}, \tilde{\mathbf{x}})_{\text{swap}(\Gamma)} \equiv (\mathbf{x}, \tilde{\mathbf{x}})$, where $\equiv$ denotes equality in distribution (and in practice, over the sample distribution); and $(ii)$ $\tilde{\mathbf{x}}$ is independent of $y$ given $\mathbf{x}$ (which is trivially met by ignoring $y$ during the design of the knockoffs).

After knockoff samples are created for the VS training dataset, the framework relies on the computation of *variable statistics* $v_n = v_n((\mathbf{X}, \tilde{\mathbf{X}}), \mathbf{Y})$ for each predictor variable that may depend on all the original variables, all the knockoff variables, and the label; here $\mathbf{X}$, $\tilde{\mathbf{X}}$ and $\mathbf{Y}$ denote the set of samples of the original and knockoff variables and the corresponding labels, respectively. The variable statistics generator must have a *flip-sign* property: the sign of the variable must change when the corresponding entries of the vectors are swapped, i.e., $v_n((\mathbf{X}, \tilde{\mathbf{X}})_{\text{swap}(n)}, \mathbf{Y}) = -v_n((\mathbf{X}, \tilde{\mathbf{X}}), \mathbf{Y})$ for each $n$. Flip-sign variable statistics can be easily obtained by creating variable *importance scores* $(\mathbf{s}, \tilde{\mathbf{s}}) = (\{s_1, \ldots, s_N\}, \{\tilde{s}_1, \ldots, \tilde{s}_N\}) = t((\mathbf{X}, \tilde{\mathbf{X}}), \mathbf{Y})$, measuring the impor-

tance of the corresponding variables for the set of sample labels $\mathbf{Y}$, that obey the swap property $t((\mathbf{X}, \hat{\mathbf{X}})_{\mathrm{swap}(n)}, \mathbf{Y}) = (\mathbf{s}, \tilde{\mathbf{s}})_{\mathrm{swap}(n)}$ for each $n$; we can then define $v_n = s_n - \tilde{s}_n$. VS proceeds by finding a magnitude threshold $\tau$ over the values of $\mathbf{v} = [v_1, \ldots v_N]$ that yields a sufficiently small ratio $\hat{q} = |\{n : v_n < -\tau\}|/|\{n : v_n > \tau\}|$ (e.g., the ratio between the number of indices with negative evidence and the number of indices with positive evidence), since $\hat{q}$ gives a proxy for the target FDR $q$ (Barber & Candès, 2019; Candès et al., 2018).

The promising recent development of model-X knockoffs (Candès et al., 2018) simplifies the original framework above by showing that knockoffs only require distribution invariance to swapping on a single variable basis. In other words, knockoffs can be constructed simply by sampling each knockoff variable according to the conditional likelihood of the original variable given the rest of the original variables and the previously obtained knockoff variables, $p(x_n|\tilde{\mathbf{x}}_{1:n-1}, \mathbf{x}_{-n})$; therefore, each knockoff variable can be generated independently in a particular sequence. The resulting simplicity of the model-X knockoff framework provides an exciting opportunity for the use of modern data models that aim to learn the distribution of the data $\mathbf{x}$ in novel schemes for knockoff generation.

An **autoencoder** is a neural network that aims to obtain a low-dimensional data representation by ($i$) setting both the input and output of the network to be the data $\mathbf{x}$ itself, so that the network learns to recreate the data based on any of the hidden layer representations; and ($ii$) including a specific hidden layer of low dimensionality whose output activation $\mathbf{z}$ provides a compact representation from which the data can be recovered. The neural network is thus naturally divided into encoder and decoder portions. Autoencoders have been used in ML to provide compact data representations $\mathbf{z}$ that are then fed into modern ML algorithms. However, the mapping obtained by the neural network is deterministic, and does not provide information on the distribution of the data $p(\mathbf{x})$. Variational autoencoders (VAEs) inherit this name because they follow the structure mentioned above. However, the low-dimensional representation $\mathbf{z}$ for the data follows a prescribed probability distribution $p(\mathbf{z})$. Consequently, the encoder and decoder aim to learn the forward and backward data distributions $p(\mathbf{z}|\mathbf{x})$ and $p(\mathbf{x}|\mathbf{z})$ (Doersch, 2016).

More specifically, we assume that there exists a data generating function $\mathbf{x} = f(\mathbf{z}; \theta) : \mathcal{Z} \times \Theta \to \mathcal{X}$ that provides a map between a parameter value $\mathbf{z}$ and the expected signal $\mathbf{x}$ conditioned on $\mathbf{z}$, where $\theta$ denotes the various model parameters. Practical VAEs leverage the common Gaussian assumption for the conditional data and parameter distributions. Under such a probabilistic model, the VAE network aims to learn two separate functions in its two halves: the encoder learns an independent-entry multivariate Gaussian approximation $Q(\mathbf{z}|\mathbf{x}, \theta) = \mathcal{N}(\mathbf{z}|\hat{\mu}_{\mathbf{z}|\mathbf{x},\theta}, \hat{\Sigma}_{\mathbf{z}|\mathbf{x},\theta})$ for the probability distribution $P(\mathbf{z}|\mathbf{x}, \theta)$. Thus, the low-dimensional representation learned by the encoder in its narrowest layer corresponds to the approximated mean $\mu_{\mathbf{z}|\mathbf{x}} \approx \mathbf{E}[\mathbf{z}|\mathbf{x}, \theta]$ and the diagonal of the variance matrix $\hat{\Sigma}_{\mathbf{z}|\mathbf{x},\theta} \approx \mathrm{Var}(\mathbf{z}|\mathbf{x}, \theta)$. The decoder learns the mapping $\mathbf{x} = f(\mathbf{z}; \theta) = \mathbf{E}[\mathbf{x}|\mathbf{z}, \theta]$ (Doersch, 2016).

Nonlinear manifold models also aim at learning mappings from the high-dimensional space containing the data of interest to a low-dimensional representation space, which is usually described in terms of providing a parametrization of the data. As with autoencoders, the large majority of approaches are deterministic, as they aim at preserving the local geometry of the data in the high dimensional space. Furthermore, almost all existing methods for manifold learning do not provide a bidirectional mapping so that new data samples can be obtained from arbitrary representations, lacking a generative model to obtain data from arbitrary points in the parameter space.

In contrast, **Gaussian process latent variable models** (GPLVMs) (Lawrence, 2003) have been proposed to provide a probabilistic nonlinear mapping between the parameter space and the data space, inspired by probabilistic principal component analysis and kernel methods, and scalable to large-scale datasets (Hensman et al., 2013). As in principal component analysis, it is first assumed that each data sample $\mathbf{x}$ can be written as $\mathbf{x} = \mathbf{W}\mathbf{z} + \mathbf{e}$, where $\mathbf{W}$ is the basis for a subspace approximation of the data, $\mathbf{z}$ is a low-dimensional parametrization vector, and $\mathbf{e}$ is the model error. It is further assumed that $\mathbf{z} \sim \mathcal{N}(\mathbf{0}, \mathbf{I})$, that $\mathbf{e} \sim \mathcal{N}(\mathbf{0}, \beta\mathbf{I})$, and that each row of $\mathbf{W}$ is an i.i.d. zero-mean Gaussian vector with covariance $\alpha\mathbf{I}$.

Given a data set $\mathbf{X} = [\mathbf{x}_1, \mathbf{x}_2, \ldots]$, the corresponding parameter vectors $\mathbf{Z}$ can be obtained via maximum likelihood estimation with the conditional distribution that marginalizes $\mathbf{W}$:

$$p(\mathbf{X}|\mathbf{Z}, \theta) = \frac{1}{(2\pi)^{\frac{DN}{2}} |\mathbf{K}|^{\frac{D}{2}}} \exp\left(-\frac{1}{2}\mathrm{trace}(\mathbf{K}^{-1}\mathbf{X}^T\mathbf{X})\right),$$

---

**Algorithm 1** *Knockoff Construction from Arbitrary Probabilistic Generative Model (KnCAPGM)*

---

**Input:** probability distributions $p(\mathbf{x}|\mathbf{z}, \theta)$, $p(\mathbf{z}|\mathbf{x}, \theta)$, input features $\mathbf{x}$
**Output:** output knockoff features $\tilde{\mathbf{x}}$
 1: initialize knockoff $\tilde{\mathbf{x}} = \mathbf{x}$
 2: **for** $n = 1, 2, \ldots, N$ **do**
 3:     marginalize $x_n$ from the distribution $p(\mathbf{z}|\mathbf{x}, \theta)$ to obtain $p(\mathbf{z}|\mathbf{x}_{-n}, \theta)$
 4:     obtain $\mathbf{z}^*$ as a sample from $p(\mathbf{z}|\tilde{\mathbf{x}}_{-n}, \theta)$
 5:     obtain $\mathbf{x}^*$ as a sample from $p(\mathbf{x}|\mathbf{z}^*, \theta)$
 6:     update knockoff vector $\tilde{\mathbf{x}}$: $\tilde{x}_n = x_n^*$
 7: **end for**
 8: return knockoff $\tilde{\mathbf{x}}$

---

where $\mathbf{K} = \alpha \mathbf{Z}^{\mathbf{T}} \mathbf{Z} + \beta^{-1} \mathbf{I}$. In order to introduce nonlinearity in the map between $\mathbf{z}$ and $\mathbf{x}$, the matrix $\mathbf{K}$ is instead rebuilt using a kernel space representation (e.g., with a radial basis function kernel): $K_{i,j} = \kappa(\mathbf{z}_i, \mathbf{z}_j) = \alpha \exp\left(-\frac{\gamma}{2}\|\mathbf{z}_i - \mathbf{z}_j\|_2^2\right) + \beta^{-1}\delta_{i,j}$, where $i, j$ are indices of the training dataset $\mathbf{X}$. GPLVM training obtains the parameters $\theta = \{\alpha, \beta, \gamma\}$ and $\mathbf{Z}$ via an iterative method that alternatively uses maximum likelihood for $\theta$ and maximum a posteriori estimation for $\mathbf{Z}$ with the conditional likelihood of the data $\mathbf{X}$ given above, respectively (Lawrence, 2003; 2005).

## 3 KNOCKOFF FEATURE DESIGN FROM GENERATIVE MODELS

We introduce a basic algorithm for the construction of knockoff features using modern generative models from ML for rich data sources that aim to express the underlying distribution of the data. Note that in contrast to the prior work's focus on VS with Gaussian and Markov chain models, the models used here have shown to be significantly more accurate for datasets of interest, as evident by the significant performance improvements obtained in ML algorithms that leverage these models.

Our proposed algorithm leverages a generative data model that relies on the construction or estimation of two mappings between the data space $\mathbf{x} \in \mathcal{X}$ and a parametric space $\mathbf{z} \in \mathcal{Z}$, and the probability spaces for their counterparts, respectively, to express or approximate the conditional distributions $p(\mathbf{x}|\mathbf{z}, \theta)$ and $p(\mathbf{z}|\mathbf{x}, \theta)$, where $\theta$ represents the model parameters. Our procedure is inspired by the simplified framework of model-X knockoffs, which generates the knockoff sample entries $\{\tilde{\mathbf{x}}_n\}$ lexicographically by sampling from the distribution $p(x_n|\tilde{\mathbf{x}}_{1:n-1}, \mathbf{x}_{-n})$. We thus proceed iteratively over the data dimensions by $(i)$ marginalizing the parameter space distribution $p(\mathbf{z}|\mathbf{x}_{-n}, \theta)$ over the data feature dimension under consideration; $(ii)$ mapping a mixed knockoff/original feature data vector to the parameter space, e.g., drawing a sample $\mathbf{z}^*$ from $p(\mathbf{z}|[\tilde{\mathbf{x}}_{1:n-1}, \mathbf{x}_{n+1:N}], \theta)$; $(iii)$ leveraging the inverse mapping $p(\mathbf{x}|\mathbf{z}^*, \theta)$, from the parameter space to the data space, to infer the missing values for the marginalized dimension $\mathbf{x}_n$; $(iv)$ assigning the recovered data values to the knockoff feature entry $\tilde{\mathbf{x}}_n$, and preserving it for the next iteration of the algorithm. This routine is formalized as Algorithm 1; while the process described there does not take into account the values of the so-far discarded dimensions of the original data $\mathbf{x}_{1:n-1}$, we note that the generative model used to compute new knockoffs samples is trained over the original data samples, providing a dependence of the new knockoff entry value on the original data distribution and the knockoff entry values computed so far. We proceed by describing two instances of this algorithm that leverage popular generative models.

**Knockoffs from VAEs:** A VAE trained on a dataset of interest directly provides us with the relevant data probability distributions for Algorithm 1. We begin by considering the marginalization $p(\mathbf{z}|\mathbf{x}_{-n})$.

**Lemma 1.** *Assume that $\mathbf{x}|\mathbf{z} \sim \mathcal{N}(\mu_{\mathbf{x}|\mathbf{z}}, \sigma_{\mathbf{x}|\mathbf{z}}^2 \mathbf{I})$, $\mathbf{z} \sim \mathcal{N}(\mathbf{0}, \mathbf{I})$, and $\mathbf{z}|\mathbf{x} \sim \mathcal{N}(\mu_{\mathbf{z}|\mathbf{x}}, \Sigma_{\mathbf{z}|\mathbf{x}})$ (e.g., the VAE model). Then $p(\mathbf{z}|\mathbf{x}_{-n}) \propto p(\mathbf{z}|\bar{\mathbf{x}})$, where $\bar{\mathbf{x}}[m] = 0$ for $m = n$ and $\bar{\mathbf{x}}[m] = \mathbf{x}[m]$ for $m \neq n$.*

*Proof.* We begin by noting that

$$p(\mathbf{x}_{-n}|\mathbf{z}) = \frac{p(\bar{\mathbf{x}}|\mathbf{z})}{\sqrt{2\pi}\sigma_{\mathbf{x}|\mathbf{z}}\exp\left(-\frac{\mu_{\mathbf{x}|\mathbf{z}}[n]^2}{2\sigma_{\mathbf{x}|\mathbf{z}}^2}\right)} \text{ and } p(\mathbf{x}_{-n}) = \frac{p(\bar{\mathbf{x}})}{C\exp\left(\frac{\mu_{\mathbf{x}_{-n}}^T \rho_{-n} - \mu_{\mathbf{x}}[n]^2}{2\sigma_{\mathbf{x}[n]}^2}\right)},$$

Table 1: Details of datasets and experiments used in this paper.

| Dataset | Features | Samples | Classes | Type | Training set size | VAE size |
|---|---|---|---|---|---|---|
| BASEHOCK | 4862 | 1988 | 2 | Text | 994 | 1024 |
| Caltech101 (VGG19) | 4096 | 3000 | 100 | Image Features | 1000 | 1024 |
| COIL20 | 1024 | 1440 | 20 | Image | 720 | 512 |
| Cub (VGG19) | 4096 | 9750 | 195 | Image Features | 1950 | 1024 |
| Isolet | 617 | 1560 | 26 | Audio | 780 | 128 |
| MNIST | 784 | 27200 | 10 | Image | 1000 | 128 |
| PCMAC | 3289 | 1920 | 2 | Text | 960 | 1024 |
| RELATHE | 4322 | 1426 | 2 | Text | 713 | 1024 |
| Yale | 1024 | 165 | 15 | Image | 90 | 256 |

where $\rho_{-n} = \mathbb{E}[(x_n - \mu_{\mathbf{x}}[n])(\mathbf{x}_{-n} - \mu_{\mathbf{x}_{-n}})]$ and $C$ is a normalization constant. We then consider the marginalization

$$p(\mathbf{z}|\mathbf{x}_{-n}) = \frac{p(\mathbf{x}_{-n}|\mathbf{z})p(\mathbf{z})}{p(\mathbf{x}_{-n})} = \frac{p(\bar{\mathbf{x}}|\mathbf{z})p(\mathbf{z}) \cdot C \cdot \exp\left(\frac{\mu_{\mathbf{x}_{-n}}^T \rho_{-n} - \mu_{\mathbf{x}}[n]^2}{2\sigma_{\mathbf{x}[n]}^2}\right)}{p(\bar{\mathbf{x}})\sqrt{2\pi}\sigma_{\mathbf{x}|\mathbf{z}} \cdot \exp\left(-\frac{\mu_{\mathbf{x}|\mathbf{z}}[n]^2}{2\sigma_{\mathbf{x}|\mathbf{z}}^2}\right)} = C'p(\mathbf{z}|\bar{\mathbf{x}}),$$

$\square$

Lemma 1 implies that the distribution learned by the VAE already provides us with a proxy for the necessary marginal, if we set the value of $x_n = 0$ in the input of the VAE. We then proceed to obtain a sample $\mathbf{z}^* \sim p(\mathbf{z}|\mathbf{x}_{-n}, \theta)$, and use the decoder to obtain a new data sample $\mathbf{x}^*$ drawn from the distribution $p(\mathbf{x}|\mathbf{z}^*)$. We set the knockoff feature $\tilde{x}_n = x_n^*$, which is akin to "inpainting" the marginalized dimension from $\mathbf{x}$ according to the learned data distribution $p(\mathbf{x}|\mathbf{z}^*)$. Figure 1(d) illustrates the outcome of this process for several examples from three image datasets with VAE hidden layer and training dataset sizes given in Table 1.

**Knockoffs from GPLVMs:** The GPLVM provides a generative probabilistic model that gives a distribution for the data vector $\mathbf{x}$ given the value of the parameter $\mathbf{z}^*$ and the model parameters $\theta$ trained from $\mathbf{X}$. One can show that the underlying distribution is given by $p(\mathbf{x}|\mathbf{z}^*, \theta) = \mathcal{N}(\mathbf{x}|\mathbf{X}\mathbf{K}^{-1}\mathbf{k}^*, \sigma_*^2\mathbf{I})$; here $\mathbf{k}^*$ has entries $k_i^* = \kappa(\mathbf{z}_i, \mathbf{z}^*)$, where $\mathbf{z}_i$ is the low-dimensional embedding for the training sample $\mathbf{x}_i$, and $\sigma_*^2 = \mathbf{k}_i^{*T}\mathbf{K}^{-1}\mathbf{k}_i^*$. While it is difficult to compute the additional mapping $p(\mathbf{z}|\tilde{\mathbf{x}}, \theta)$ required by Algorithm 1 due to the use of the kernel, we propose a heuristic that replaces the sampling procedure by a maximum a posteriori estimate procedure based on the distribution above, i.e., we return the value of $\mathbf{z}$ that maximizes $p(\mathbf{z}|\bar{\mathbf{x}}, \theta) = p(\bar{\mathbf{x}}|\mathbf{z}, \theta)p(\mathbf{z}|\theta)/p(\bar{\mathbf{x}}|\theta)$. This procedure can be interpreted as an out-of-sample extension of the GPLVM manifold model for the point $\bar{\mathbf{x}}$, e.g., a method that provides an embedding for new (e.g., testing) data points into the low-dimensional parameter space established by the manifold. The independence assumption implicit in the conditional distribution $p(\mathbf{x}|\mathbf{z}^*, \theta)$ simplifies the marginalization required by Algorithm 1. Figure 1(e) illustrates the outcome of this process for several examples from three image datasets using a GPLVM featuring a 10-dimensional parameter space and with a training set of size given in Table 1.

## 4 KNOCKOFF-BASED FEATURE SELECTION FOR SUPERVISED LEARNING

Existing importance scores to evaluate original and knockoff variables have been limited to regression (i.e., estimation) problems. In particular, variable statistics have been developed for logistic regression (Sesia et al., 2019; Gimenez et al., 2018), linear regression via lasso and group lasso (Barber & Candès, 2015; 2019; Candès et al., 2018; Sesia et al., 2019; Gao et al., 2018; Katsevich & Sabatti, 2019), Bayesian hierarchical regression (Candès et al., 2018), and deep learning estimators (Lu et al., 2018). In contrast, our consideration of knockoffs for FS focuses on general ML problems; as a starting point, we will consider standard supervised learning.

Our feature importance scores are designed for the application of a softmax classifier, which uses class scores defined as $l_c(\mathbf{x}) = \frac{\exp(\mathbf{x}^T\mathbf{w}^c)}{\sum_{c'=1}^{C} \exp(\mathbf{x}^T\mathbf{w}^{c'})}$ for each of the classes $1, .., C$. The sample $\mathbf{x}$ is then assigned the predicted label $\hat{c}(\mathbf{x}) = \arg\max_c l_c(\mathbf{x})$; the parameter weights $\mathbf{w}^c$ are trained using a cross-entropy loss between the scores and the sample labels. We propose a simplified scoring scheme for FS, reminiscent of the concept in Lu et al. (2018), by expanding the softmax classifier class scores to an expanded input, consisting of a concatenation of the original and knockoff features:

$l_c^k(\mathbf{x}, \tilde{\mathbf{x}}) = \frac{\exp(\mathbf{x}^T \mathbf{w}^c + \tilde{\mathbf{x}}^T \tilde{\mathbf{w}}^c)}{\sum_{c'=1}^{C} \exp(\mathbf{x}^T \mathbf{w}^{c'} + \tilde{\mathbf{x}}^T \tilde{\mathbf{w}}^{c'})}$. Importance scores for each original and variable knockoff can then be obtained as the sums of squares $s_n = \sum_{c=1}^{C} (w_n^c)^2$ and $\tilde{s}_n = \sum_{c=1}^{C} (\tilde{w}_n^c)^2$, respectively. Similarly to VS, the feature statistics vector is simply obtained as the difference of the original and knockoff feature importance scores $\mathbf{v} = \mathbf{s} - \hat{\mathbf{s}}$.

## 5 RELATED WORK

There are several recent contributions to the knockoff literature for VS that are related to our work. Liu & Zheng (2018) proposed knockoff variable construction for regression based on autoencoders; however, the design of these knockoffs does not make the value of each knockoff variable independent from the original variable. In contrast, our approach is applicable to many generative models and marginalizes the feature under consideration before starting the construction of the corresponding knockoff. Jordon et al. (2019) proposed a construction for knockoff variables based on a generative adversarial network (GAN). The GAN's generative network uses the original variables and random noise as inputs and generates the corresponding knockoff variables, while its discriminative network takes as input a subset-swapped version of the concatenated original-knockoff feature vector and attempts to estimate the set of swapped variables. The approach also uses a mutual information estimate in the training objective function to promote independence between original and knockoff variables. In contrast to the data models considered in this paper, GANs do not aim to obtain a low-dimensional parameterization of the data or its distribution; furthermore, we are not aware of an implementation of this method being available for comparison.

More recently, Romano et al. (2019) proposed the design of knockoffs based on a deep learning network that is also inspired by the GAN architecture. Similarly to the original Gaussian knockoff constructions of Barber & Candès (2015; 2019); Candès et al. (2018), the formulation of the objective function for their proposed algorithm is still centered on the matching of moments between the distributions for the original and knockoff variables. We compare the performance of this approach with our own in the sequel.

Separately, Lu et al. (2018) proposed a framework for the design of variable statistics for estimation based on a deep learning network. The statistics are obtained by augmenting the deep learning network with a pair of pre-processing layers, featuring an input of original and knockoff variables, that merge each pair of matching variables (original/knockoff) at the first layer, and assigns trainable weights to each of them; the output of this layer is then mapped directly to the input of the original deep learning network. This straightforward approach allows for easy implementation of variable importance scores in cases where the first action on the input features is a linear projection, as is the case with the softmax classifier leveraged in our experiments.

## 6 EXPERIMENTAL RESULTS

To test the performance of our proposed knockoff designs, we consider several datasets commonly used as benchmarks for FS in ML (Li et al., 2017; 2019), including image, test, and audio data, under a softmax classifier setup. Detailed properties of those datasets are summarized in Table 1. We combine the variable relevance scores of Section 4 for the aforementioned softmax classifier with the two knockoff construction schemes proposed in Section 3, as well as with two previously proposed and available constructions: second order/Gaussian approximation (Candès et al., 2018) and deep knockoffs (Romano et al., 2019). The knockoff construction schemes proposed in this paper were implemented in Python using the Keras and GPy toolboxes (GPy, 2019); we used the implementations available from (Sesia & Romano, 2019) for the two existing constructions tested here. Additionally, we tested the performance of several FS approaches from the literature (Duda et al., 2001; Robnik-Šikonja & Kononenko, 2003; Wright, 1965; Nie et al., 2010), using implementations available from the scikit-feature toolbox (Li et al., 2017; Li & Cheng, 2019). The datasets were randomly split in half into training and testing sets, except in cases where equal representation of classes requires a smaller training set, or when the majority of the FS algorithm tests lasted more than 30 CPU-days; see Table 1 for details.

For knockoff construction, we implemented VAEs with one hidden layer for both the encoder and decoder of the sizes indicated in Table 1 for each dataset; the low-dimensional representation $z$ is five-dimensional for all datasets as well. The VAE nodes use the ReLU activation function; for the VAE, datasets are normalized to be in the range [0,1]. We also implemented GPLVMs with parameter space dimension 10 for all datasets. The latent dimensionalities of the models tested were selected by evaluation of the quality of the reconstructions obtained from the original data.

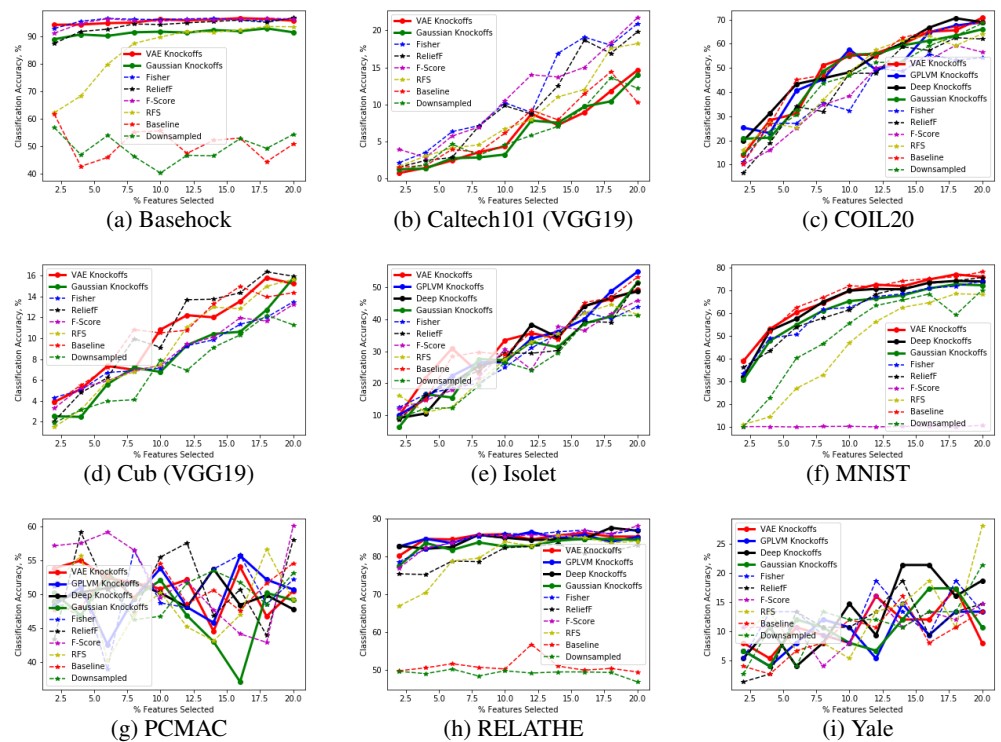

Figure 2: *Performance of a deep learning softmax classifier for multiple datasets as a function of the percentage of features selected by multiple baseline and knockoff-based algorithms. The algorithms tested for comparison are Gaussian knockoffs (Candès et al., 2018), deep knockoffs (Romano et al., 2019), Fisher score (Duda et al., 2001), ReliefF (Robnik-Šikonja & Kononenko, 2003), F-Score (Wright, 1965) and RFS (Nie et al., 2010), For datasets featuring rich training data, knockoff-based algorithms outperform baselines, with Gaussian knockoffs outperformed by their counterparts.*

We evaluate the performance of the same softmax classifier trained and tested with the features selected by each of the FS and VS frameworks using the score of Section 4, where the percentage of selected features varies from 2% to 20%. Figure 2 shows that the proposed knockoff-based FS methods outperform the baseline approaches in almost all instances. Note that results are missing for some of the baseline methods because the relevant simulation did not complete after an upper limit of 30 CPU-days. We see that VAE, GPLVM, and deep knockoffs often provide better performance than standard Gaussian knockoffs. In some cases, the number of samples available to fit the data models used in knockoff generation is not sufficient to obtain good performance for knockoff-based algorithms. We also conjecture that the performance of the knockoff-based models studied is dependent on the suitability of the underlying generative model for the dataset being considered. As a sanity check, we tested the performance of uniform downsampling of the feature vector, as well as the knockoff-based FS method where all knockoff feature scores have zero value (e.g., knockoffs are not considered during feature selection), which we dub *baseline* in Figure 2.

## 7    DISCUSSION AND CONCLUSIONS

In this paper, we have proposed an algorithm for FS based on the knockoff variable framework. Our goal is to extend its applicability to general problems in ML, and we focused on the specific problem of classification using softmax activations. We have observed that the performance of FS is dependent on the suitability of the model used in knockoff design for the data being considered and for the number of data points available for training. Given that the data models used by our approach (and by deep knockoffs) are better suited for many modern data-centric problems, we obtain better performance using them vs. the original Gaussian model.

There is additional potential for the knockoffs described here. The fact that knockoff generation does not require knowledge of the data labels (and their joint distribution with the data) implies that knockoffs can be generated for data from unsupervised learning problems. Existing scores for unsupervised FS could be used on knockoff features as well.

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
