# OpenReview forum: "Knockoff-Inspired Feature Selection via Generative Models"
_ICLR.cc/2020/Conference — Reject_

### Official Review · AnonReviewer3 · 2019-10-14
**Official Blind Review #3**

**Rating:** 3

**Review:**

This paper proposes a new feature selection method by integrating the knockoff procedure and generative models.
The paper is clearly written and easy to read. However, I have the following concerns:

- The motivation is not clear. The advantage of the knockoff procedure is that it can find relevant features (variables) with statistical guarantees such as FDRs, which is clearly discussed in the paper.
    However, the objective of this paper is to design a better feature selection method for prediction, where the statistical guarantee is usually not important.
    Hence the advantage of using the knockoff procedure is not clear.
- In addition to the above issue, the empirical performance of the proposed method is not convincing.
    In experiments, Figure 2 shows that the proposed approach does not have significant advantage compared to existing methods.
    It seems that this is a natural consequence as the knockoff procedure is not designed for feature selection.
    Thus, in its current state, the advantage of the proposed approach is not well presented.
- Also, the proposed approach is not theoretically analyzed.
    Since the proposed method is based on the knockoff procedure, it would be interesting if there is some statistical guarantee for the selected features.

Minor comments:
- P.5, L.-4: "SInce" -> "Since"


**Experience Assessment:**

I have published one or two papers in this area.

**Review Assessment: Checking Correctness Of Derivations And Theory:**

I assessed the sensibility of the derivations and theory.

**Review Assessment: Checking Correctness Of Experiments:**

I carefully checked the experiments.

**Review Assessment: Thoroughness In Paper Reading:**

I read the paper at least twice and used my best judgement in assessing the paper.

---

> ### Author Response · Authors · 2019-11-10
> **Response to Blind Review #3**
>
> Thank you for your thoughtful comments and careful reading of the manuscript. We have posted a revision to address some of your questions and provide some responses below (matching the order of the comments).
>
> We agree with the reviewer that the statistical guarantees from the original knockoff variable framework from variable selection does not translate to feature selection, because in machine learning it is seldom the case that the label is independent from a large set of entries of the input vector. Nonetheless, we believe that there is a contribution in applying the knockoff framework to the long-standing feature selection problem in machine learning.
>
> We relied on datasets and algorithms that have been used in the feature selection literature for the numerical comparison - see the references by J. Li et al. It is apparent that some of these datasets provide significant challenges to all tested feature selection methods, and it is not surprising to us that in some cases one cannot perform feature selection without undergoing significant loss in performance.
>
> The typo has been corrected.

---

### Official Review · AnonReviewer1 · 2019-10-23
**Official Blind Review #1**

**Rating:** 3

**Review:**

This paper presents an interesting use of a knockoff framework similar to that of model-X knockoffs (Candes et al., 2018) in generative models like VAE and GPLVM for feature selection in supervised learning.

On the flip side, it is really hard to tell from the experimental results (Fig. 2) what performance benefits their proposed designs bring over the state of the art. I would encourage the authors to provide a more detailed analysis of the conditions under which their proposed designs would outperform the state of the art (or not). The author should consider presenting their results through other means that can better highlight the performance benefits of their proposed designs.

Besides applying their knockoff designs to a softmax classifier, can the authors provide examples of applications to other supervised learning models?


Proof of Lemma 1: Without further assumptions from that stated in the lemma, can the authors provide a derivation (or result that they've used) and explanation for the mean vector and covariance matrix of the joint probability p(z,x_n|x_{−n}) = p(z|x)p(x_n|x_{−n})? In particular, why are z and x_n conditionally independent given x_{-n}, as reflected in their zero covariance value?


Page 2: The authors say that "they must be independent from the labels to be predicted". Shouldn't it be conditionally independent (2nd paragraph, page 3) instead?



Minor issues
Page 5: The notation of tilde{bold{x}}_n is ambiguous since the subscript is used to index the data sample (page 2). Is it intended to be without a bold?

Step 3 of Algorithm 1: I would have preferred that it is stated consistently with that in the main text (step i in 2nd paragraph of Section 3).

Page 5: This sentence is difficult to parse: "the training dataset for the generative models used to compute new knockoffs sample values is trained over the original data samples".

Page 5: SInce?

Page 6: I cannot find Figure 1(g) and Figure 1(h).

**Experience Assessment:**

I have read many papers in this area.

**Review Assessment: Checking Correctness Of Derivations And Theory:**

I carefully checked the derivations and theory.

**Review Assessment: Checking Correctness Of Experiments:**

I carefully checked the experiments.

**Review Assessment: Thoroughness In Paper Reading:**

I read the paper thoroughly.

---

> ### Author Response · Authors · 2019-11-10
> **Response to Blind Review #2**
>
> Thank you for your thoughtful comments and careful reading of the manuscript. We have posted a revision to address some of your questions and provide some responses below (matching the order of the comments).
>
> We relied on datasets and algorithms that have been used in the feature selection literature for the numerical comparison - see the references by J. Li et al.
>
> The knockoffs used here can be also used with relevance metrics defined elsewhere for regression and estimation. Furthermore, the DeepPINK relevance scores by Lu et al. (2018) can be adapted to other applications of deep networks in supervised learning. We will attempt to collect results for the same datasets and feature selection schemes using a deep learning classifier with the DeepPINK scores, and will provide updated results (e.g., in an appendix) if results are available by the end of the discussion period.
>
> The probability statement questioned has been removed as we have changed Lemma 1, but it can be simplified to $p(a,b|c) = p(a|b,c) \cdot p(b|c)$.
>
> Regarding the independence between knockoff and variables described in p. 2, we make a correction to “the knockoffs must be independent from the labels to be predicted given the original variables.”
>
> We slightly changed the description of Step 3 in Algorithm 1, as well as the referenced sentence in Page 5.
>
> The notation $\tilde{\mathbf{x}}_n$ refers to the knockoff vector for the sample $\mathbf{x}_n$. Non-bold would refer to an entry of a vector $\mathbf{x}$; we use $\tilde{x}_{n,j}$ for entries of the former vector. Typos have been corrected. References to Figs. 1(g-h) have been corrected to Figs. 1(d-e).

---

### Official Review · AnonReviewer2 · 2019-10-27
**Official Blind Review #2**

**Rating:** 3

**Review:**

This paper proposes a way of employing modern generative models like VAEs and GPLVMs within the recently popular “knockoff” framework for variable/feature selection. Roughly speaking, the main idea of knockoffs is to construct a duplicate (“knockoff”) of each regression variable which matches its distributional properties but is independent of the response variable when conditioning on the true input variable the knockoff is based on. Under certain assumptions, fitting a model on top of both original and knocked-off features and observing the difference between the fitted coefficients for the true and the knockoff features can then be used for variable selection with guaranteed false discovery rate. The authors of this paper suggest that many of the current methods for construction of knockoffs may not be appropriate for image, text, and other types of datasets commonly used in modern ML, and suggest a heuristic way of employing VAEs and GPLVMs for this purpose. The paper concludes with an empirical study which shows that their algorithm is competitive with existing feature selections methods in terms of post-selection accuracy of the fitted model.

I am currently leaning towards recommending rejection. While the paper is nicely written and does a good job of reviewing knockoffs, I see two main issues: (1) I am not sure about applications for the proposed algorithm; in particular, the authors allude to use on devices with limited memory and computational power, but do not discuss why the Johnson-Lindenstrauss transform or some of the many low-precision implementations of neural networks (binarised neural networks, xnor-nets, …) cannot be used; furthermore, in the case of images, I am not sure why there is no comparison to simple downsampling to a smaller resolution; (2) The reported results do not show a reliable improvements over existing methods.


Major comments:

- I am not entirely sure lemma 1 is correct. In particular, mu_{z | x} tends to be a function of the whole vector x including x_n. For example, if (a, b) are jointly distributed according to a bivariate normal, then E (b | a) = E(b) + Sigma_{ab} Sigma_{aa}^{-1} (a - E(a)) where Sigma is the corresponding covariance matrix. Hence claiming that the marginal distribution z | x_{-n} is N ( mu_{z|x} , Sigma_{z | x} ) seems wrong as mu_{z | x} will generally depend on x_n (similarly to how E(b | a) depends on “a” above) which cannot be the case when x_n is marginalised. If z depends linearly on x, there is a standard expression for the distribution you seek. However, with the non-linear dependence employed, e.g., within VAE, working out a closed form expression may be quite a challenge. Can you please clarify or drop this result from your paper if it indeed turns out incorrect?

- Can you please clarify why don’t you benchmark against the cited algorithm proposed in Lu et al. (2018)?

- Can you please explain how was the latent dimensionality for the VAEs (5) and GPLVMs (10) selected? Also, for the algorithms where random seed plays a role (e.g., VAEs), how many random seeds were used (are the numbers reported in fig.2 a result of averaging over multiple seeds)? Relatedly, have you tested whether starting from different seeds results in the same subset of variables selected (e.g., when using VAEs)?


Minor comments:

- In the 1st sentence of the introduction, “prevalence” can be high or low, increase or decrease, etc. but “becoming increasingly pervasive” does not sound right. Please consider rewording.

- In fig.1e, do you know why GPLVM leads to such a significant mode collapse?

- Just after the 1st display on p.5, “SInce” -> “Since”

**Experience Assessment:**

I do not know much about this area.

**Review Assessment: Checking Correctness Of Derivations And Theory:**

I assessed the sensibility of the derivations and theory.

**Review Assessment: Checking Correctness Of Experiments:**

I assessed the sensibility of the experiments.

**Review Assessment: Thoroughness In Paper Reading:**

I read the paper at least twice and used my best judgement in assessing the paper.

---

> ### Author Response · Authors · 2019-11-10
> **Response to Blind Review 1**
>
> Thank you for your thoughtful comments and careful reading of the manuscript. We have posted a revision to address some of your questions and provide some responses below (matching the order of the comments).
>
> There are cases where selecting dimensions of the original data, rather than processing them, is desirable - e.g., if the dimensions of the data have semantic meaning or the application benefits from explainability in the decision making process.
>
> Regarding images (and other signals), we do agree it would make sense to provide a comparison with downsampled input vectors. Thus, we have tested regular-interval downsampling as an alternative to data-dependent feature selection.
>
> Thanks for pointing out the mistake in Lemma 1. We have changed the lemma to better motivate the choice of marginalization in the VAE.
>
> Note that Lu et al. (2018) does not introduce a knockoff variable construction, but rather introduces a relevance score designed for deep learning-based estimation. The score proposed here for a softmax classifier is similar in spirit to that of Lu et al.
>
> The latent dimensionalities of the models tested were selected by evaluation of the quality of the reconstructions obtained from the original data. We have added this information to the manuscript.
>
> In the first sentence of the introduction, we replace “prevalence” by “availability”. We also corrected typos.
>
> We conjecture that GPLVM requires a more dense sampling of the assumed data manifold in order to provide good generalizations for some of the datasets used.

---

### Decision · Program_Chairs · 2019-12-19

**Decision:**

Reject

**Comment:**

This manuscript proposes feature selection inspired by knockoffs, where the generative models are implemented using modern deep generative techniques. The resulting procedure is evaluated in a variety of empirical settings and shown to improve performance.

The reviewers and AC agree that the problem studied is timely and interesting, as knockoffs combined with generative models have recently shown promise for inferential problems. However, the reviewers were unconvinced about the motivation of the work, and the strength of the empirical evaluation results. In the option of the AC, this work might be improved by focusing (both conceptually and empirically) on applications where inferential variable selection is most relevant e.g. causal settings, healthcare applications, and so on.